# Toward a Mechanism-Driven Integrated Framework to Link Human Exposure to Multiple Toxic Metal(loid) Species with Environmental Diseases

**DOI:** 10.3390/ijms25063393

**Published:** 2024-03-16

**Authors:** Jürgen Gailer

**Affiliations:** Department of Chemistry, University of Calgary, 2500 University Drive NW, Calgary, AB T2N 1N4, Canada; jgailer@ucalgary.ca

**Keywords:** arsenic, mercury, methylmercury, cadmium, blood plasma, red blood cells, organs, bioinorganic chemistry, integration, mechanism of chronic toxicity

## Abstract

The ongoing anthropogenic pollution of the biosphere with As, Cd, Hg and Pb will inevitably result in an increased influx of their corresponding toxic metal(loid) species into the bloodstream of human populations, including children and pregnant women. To delineate whether the measurable concentrations of these inorganic pollutants in the bloodstream are tolerable or implicated in the onset of environmental diseases urgently requires new insight into their dynamic bioinorganic chemistry in the bloodstream–organ system. Owing to the human exposure to multiple toxic metal(loid) species, the mechanism of chronic toxicity of each of these needs to be integrated into a framework to better define the underlying exposure–disease relationship. Accordingly, this review highlights some recent advances into the bioinorganic chemistry of the Cd^2+^, Hg^2+^ and CH_3_Hg^+^ in blood plasma, red blood cells and target organs and provides a first glimpse of their emerging mechanisms of chronic toxicity. Although many important knowledge gaps remain, it is essential to design experiments with the intent of refining these mechanisms to eventually establish a framework that may allow us to causally link the cumulative exposure of human populations to multiple toxic metal(loid) species with environmental diseases of unknown etiology that do not appear to have a genetic origin. Thus, researchers from a variety of scientific disciplines need to contribute to this interdisciplinary effort to rationally address this public health threat which may require the implementation of stronger regulatory requirements to improve planetary and human health, which are fundamentally intertwined.

## 1. Introduction

Every organism has evolved an intricate biomolecular network to orchestrate the absorption of sufficient amounts of essential elements from the environment to maintain a healthy life [1]. These homeostatic regulation processes avoid the scrambling of metal ions after the ingestion of food items containing a large number of essential metal ions and direct them by exquisitely controlled mechanisms to the sites where they are needed for the assembly of metalloproteins [2]. Thus, deficiency or an excess of essential elements in tissues is avoided to ascertain the stability of the organism’s internal environment in response to fluctuations in external environmental conditions [3,4]. The earth’s crust, however, also contains potentially toxic metal(loid)s, which have been increasingly mobilized by anthropogenic activities from the geosphere into the biosphere [5] and the hydrosphere [6] since the onset of the industrial revolution in the 1760s. Accordingly, the ‘flux’ of these potentially toxic elements ‘through’ the gastrointestinal (GI) tract of all organisms has been gradually increasing with ramifications on human health that remain poorly defined to this day [3,7].

The first adverse health effects that were triggered by the exposure of humans to toxic metal(loid)s were neuropsychiatric and neurodevelopmental disorders in workers who extracted these elements from ores for the production of consumer goods on an ‘industrial’ scale (e.g., Hg to make barometers and Pb to produce water pipes) [5]. Occupational exposure, however, remained largely unrecognized by the general public as comparatively few people were affected. The fact that environmental exposure to toxic metal(loid)s, predominantly via drinking water and contaminated food, can also severely affect human health was brought to public attention by several pollution disasters. The environmental exposure of residents who lived in the vicinity of industrial mining or production sites in Japan to Cd^2+^ and CH_3_Hg^+^ severely affected their health [8]. Relatedly, the drilling of wells to provide ‘clean drinking water’ in West Bengal/India and Bangladesh inadvertently mobilized geogenic arsenite (As^III^) into the drinking water, thus causing adverse health effects that affect millions of people [5]. In a similar vein, the more convenient delivery of drinking water into households via Pb pipes liberated Pb^2+^ into the drinking water, which caused kidney damage as well as muscle and joint pain in millions of people [5].

While these pollution disasters propelled As, Cd, Hg and Pb to be recognized as the big four elements of environmental concern, the general population today still mostly associates environmental pollution with the lack of vegetation growth in the affected areas [9], without realizing that the responsible pollutants also pose a threat to biodiversity and human health [10], provided there is an exposure pathway. A study that addressed the effect of global environmental pollution on human health in 2015 estimated that 9 million people died of direct or indirect pollution related causes [11,12], which likely represents an underestimation as the adverse effects of many environmental contaminants are not well understood [10]. Since the chronic exposure of human populations to a variety of environmental pollutants also significantly affects reproductive health in certain regions [13], environmental pollution has de facto become a planetary threat that requires a paradigm shift about how we manage public health [14]. After all, the average life expectancy of Americans began to decline for the first time in 2014 for three consecutive years [15], with pollution being a contributing factor [16]. Thus, business as usual is a strategy that is incompatible for a species that needs to be a responsible steward for the planet we inhabit [17].

Among the many environmental pollutants that are present in the biosphere, toxic metal(loid)s are in a ‘league of their own’. Firstly, they cannot be degraded once released to the environment and, therefore, tend to ‘build up’ in ecosystems or remain therein for eons after contamination has ceased [18]. Secondly, chronic human exposure to exceedingly small doses of toxic metal(loid) species is associated with severe adverse health effects, which is why the maximal tolerable daily consumption of As, Cd, Hg and Pb ranges between 2 and 250 μg/day [19]. Direct experimental evidence in support of the exposure of the average population to the aforementioned metal(loid)s is readily available from biomonitoring studies, which involve the accurate quantification of these elements in the human bloodstream [20]. While the average blood concentrations of As, Cd, Hg and Pb for adults and children reflect the fact that these elements are naturally present in the earth’s crust [21], it is impossible to state with confidence if these concentrations are problematic from a public health point of view [22], particularly with regard to children [23] and individuals that may be more susceptible than others [10]. Disconcertingly, blood transfusions are becoming a source of heavy metal poisoning in some developing nations as screening is routinely done for various blood-borne pathogens, but not for toxic metal(loid) levels [24].

In terms of exposure sources, one needs to consider the inhalation of contaminated air, which contains toxic metal(loid) species adsorbed onto particulate matter (PM), such as PM_10_ and PM_2.5_ [25], the ingestion of contaminated food [26,27] and/or drinking water [28] as well as the inadvertent exposure from consumer products, such as inexpensive jewelry [29], cosmetics [30,31] or alternative medicines [31]. While the inhalation of PM_10_/PM_2.5_-laden air was a minor contributor to the total human body burden of toxic metal(loid)s in the past, it is rapidly becoming more important as 56% of the world’s population live in megacities today. Thus, humans populations are increasingly exposed to PM_10_ and PM_2.5_ over their lifetime [16,32,33,34]. Another exposure source that increasingly contributes to the pulmonary exposure to toxic metal(loid)-laden PM_10_ and PM_2.5_ is the climate change-related increased frequency of wildfires in certain parts of the world (e.g., in California), which involves the inhalation of wildfire smoke and dust-borne metals, including hexavalent Cr [35] and Hg [36,37]. By far, the most important environmental exposure source for toxic As, Cd, Hg and Pb species for the general population is the ingestion of contaminated food and drinking water [5]. With regard to the former, the ongoing contamination of agricultural soils with potentially toxic metal(loid)s threatens food safety globally [38] and requires increased research efforts to develop effective bioremediation strategies [26,39,40]. The exposure to toxic metal(loid)-contaminated food is also driven by climate change-induced drought mitigation efforts which are forcing many affected jurisdictions to use toxic metal(loid) species laden wastewater for the irrigation of food crops, thereby hastening the contamination of the food chain [41]. Ground water remains a considerable exposure source for human populations to toxic metal(loid) species [42,43,44], as 2.5 billion people globally rely on this source to generate drinking water [28], but regular testing is often missing [45].

Based on the increasingly important role that metals will continue to play in our society for the manufacturing of electronic and other high-tech devices [46,47] and being faced with regulatory deficiencies in terms of protecting public health from the effect of the mining for metals [48] and other pollutants [14], the fraction of the global population that is environmentally exposed to potentially toxic metal(loid) species is destined to increase. Accordingly, increased research efforts should be directed at determining precisely how many people’s health is adversely affected globally and to identify preventable risk factors to initiate rational public health policy-based interventions to improve the quality of life for all [17,49]. Considering that the human genome project was completed in 2000, which revealed that the ‘genetics parts list seems insufficient to account for the origin of the numerous grievous illnesses’ [50], we urgently need to establish causal links between the exposure to inorganic pollutants and grievous diseases to tackle this massive exposure–response problem [10,51], which is easier said than done.

Owing to the biological compartmentalization of humans into ~200 cell types, the associated biological complexity and the small concentrations of toxic metal(loid)s that are associated with adverse health effects, it is not surprising that our understanding of their biochemical transformations in the bloodstream–organ system remains incomplete. One promising research strategy to tackle these knowledge gaps is to unravel the biotransformations of individual toxic metal(loid) species in biological compartments in the order in which nutrients and, thus, chemical matter more generally flow through organisms, namely plasma, red blood cells and organ cells to establish a conceptual framework of their chronic toxicity in the GI-tract–bloodstream–organ system (Figure 1). A mechanism-driven integrated approach holds promise to establish a framework which may allow one to causally link human exposure to inorganic pollutants and environmental diseases, including type 2 diabetes [52] and autism [53] as well as chronic kidney disease (CKD) [54,55], whose incidence and prevalence are currently increasing at a disconcerting rate. In addition, the etiology of at least 8 other grievous human diseases, including Alzheimer’s disease, multiple sclerosis and Parkinson’s disease remains elusive and may be directly or indirectly caused by chronic exposure to environmental pollutants [50].

This review aims to address the arguably biggest problem in the postgenomic era of causally linking human exposure to specific inorganic pollutants that are contained in air [58], drinking water [59] and soil [39] with the onset of more non-communicable environmental diseases than we are currently aware of. Owing to this broad scope, it is impossible to cite all relevant contributions from those who work in this area and this review instead relies on the arguably subjective choice of research and review articles to illustrate important concepts. Furthermore, it is attempted to keep technical details about how results were obtained to a minimum, unless they are needed for context to focus more on the toxicologically important mechanisms. The review itself is structured into five parts. The first one aims to answer the conceptual question as to whether the proposition that the chronic exposure to toxic metal(loid)s is causally involved in the etiology of more environmental disease that we currently know using purely bioinorganic chemistry arguments and taking into account that the effects of toxic metal(loid) species on cells were started to be directly linked with individual molecular targets only within the last two decades or so [60,61]. It will be argued that the application of systems toxicology [62] and metallomics approaches [56,63] have provided, and will continue to provide, much needed insight. Part two will give a succinct summary of recent advances into the toxicological chemistry of Hg^2+^, CH_3_Hg^+^ and Cd^2+^ in blood plasma, while part three will provide an equivalent update on their bioinorganic chemistry within red blood cells (RBCs). In part four, the focus will be on highlighting recent advances into the bioinorganic chemistry of the toxic metal(loid) species of interest with toxicologically relevant mechanisms of action that unfold in target organs. Part five will illustrate the concept of a mechanism-driven framework to link the chronic exposure of humans to toxic metal(loid) species with environmental diseases. While governmental policies that regulate toxic metal(loid)s in the environment will not be covered, the interested reader is referred to a recent blook chapter which provides a brief overview of the risk assessment framework in different jurisdictions, namely in Canada, the United States and the European Union [64]. Throughout this review, open questions are highlighted in areas in which progress is needed to define a better path toward establishing a framework to causally link exposure to toxic metal(loid) species with environmental diseases.

## 2. Is Human Exposure to Toxic Metal(loid) Species Linked to the Etiology of Non-Communicable Environmental Diseases?

An environmental disease can be defined as an abnormality/disorder of function in organs of the body that has a known cause and results in a distinctive group of symptoms or anatomical changes, which can be brought about either by a chemical agent (the inhalation of volatile Cd species causes metal fume fever [65]), or a pathogenic organism, such asClostridium botulinum which secretes the botulinum toxin that causes ‘botulism’. To this end, exposure to each of the big four metal(loid) species of environmental concern causes a rather distinct breakdown of homeostatic control mechanisms resulting in highly specific symptoms. Pb^2+^ poisoning, for example, is referred to as ‘plumbism’ (from ‘plumbum’, which is latin for lead) and the intoxication with As^III^ (e.g., from drinking water) as arsenicosis. The neurological symptoms that result from the exposure to CH_3_Hg^+^ is called Minamata disease and the intoxication with Cd^2+^ as ‘itai-itai’ disease, which is japanese for ‘ouch-ouch’. These examples conclusively demonstrate that the exposure to either As^III^, Cd^2+^, CH_3_Hg^+^ or Pb^2+^ will cause non-communicable environmental diseases which are the result of unique dyshomeostasis phenomena in the body. Is it therefore conceivable that chronic human exposure to the aforementioned non-essential element species may also be involved in inducing dyshomeostasis phenomena that present in individuals as type 2 diabetes, autism, chronic kidney disease or any of the other 11 diseases of unknown etiology? To critically assess this proposition, six conceptual arguments can be identified (Table 1).

Combined with differences in the genetic makeup of human individuals, the proposition that chronic human exposure to inorganic pollutants disrupts specific homeostatic regulation processes that may be directly or indirectly implicated in the etiology of more environmental diseases than we currently think [50,75] is reasonable, but also inherently difficult to unravel considering the exceedingly complex interplay that unfolds after inorganic environmental pollutants infiltrate a biochemically complex system that is intricately tied to the genetic susceptibility of individuals [56]. Since it is likely that more environmental diseases than we are aware of are caused by the chronic exposure of human populations to toxic metal(loid) species, this approach also holds inherent transformative potential for the development of more precise and effective disease prevention approaches to reduce the global burden of disease [17], thus offering inherent economic benefits [76].

## 3. Advances in the Toxicological Chemistry of Metal(loid)s in Blood Plasma

Studies into the toxicological chemistry of Cd^2+^, CH_3_Hg^+^ and Hg^2+^ in plasma have focused on identifying and structurally characterizing the binding sites of the corresponding plasma binding proteins and to determine the associated thermodynamics of these interactions. While the two most important binding proteins for Cd^2+^ in plasma are α_2_macroglobulin and human serum albumin (HSA) [77], there is considerable evidence for an additional plasma protein, namely apo-transferrin and another eight plasma proteins for which there is some evidence for Cd-binding [78]. With regard to CH_3_Hg^+^, there is direct evidence for its binding to the Cys-34 of HSA [79], which is therefore also the binding site on rabbit serum albumin (RSA) at near-physiological conditions [80]. In contrast, there is little direct experimental evidence for the binding of Hg^2+^ to Cys-34 in HSA [77,81]. With the intent of integrating the bioinorganic chemistry of toxic metal(loid) species that unfolds in the bloodstream with processes in organs, however, it is critical to elucidate the conditions at which they are mobilized from their respective plasma proteins and to structurally characterize the formed metabolites as these will impinge on target organ(s) to determine the toxicological effects.

In this context, one needs to appreciate that blood not only contains thousands of plasma proteins, but also ~100 mM of chloride as well as >400 small molecular weight (SMW) metabolites [82], including L-cysteine (Cys) and L-homocysteine (hCys) [83]. Owing to the fact that Cd^2+^, CH_3_Hg^+^ and Hg^2+^ have a strong affinity for thiols [2] and considering that the Cys concentration in human plasma is ~10 μM and that of the corresponding hCys concentration is 15–250 μM, respectively (15 μM in healthy adults and up to 250 μM in hyperhomocysteinemia patients) [84], it is not surprising that in vivo studies using rats demonstrated Cys to be involved in the translocation of Cd^2+^ [85] and Hg^2+^ [86] to the kidneys and in the short-term distribution of CH_3_Hg^+^ to the kidneys, the liver and the cerebrum [87]. Direct experimental evidence for the formation of toxic metal(loid)–SMW thiol complexes in the presence of 100 mM Cl^−^ and their structural characterization, however, has not been reported even though they likely play an important role in the metal trafficking between their respective protein binding sites en route to the uptake mechanisms located at the surface of target organ cells.

Recent studies have therefore investigated the interaction of the aforementioned SMW thiols with rabbit serum albumin (RSA) which contained bound CH_3_Hg^+^ as well as human serum albumin (has) which contained equimolar Hg^2+^ and Cd^2+^ by liquid chromatography using a PBS buffer mobile phase (pH 7.4) in conjunction with size-exclusion chromatography (SEC). In addition, the formation of toxicologically relevant Cd complexes was investigated using anion-exchange chromatography (AEX) in conjunction with a 100 mM NaCl/5 mM Tris buffer mobile phase (pH 7.4). Observing the retention time of Cd as a function of increasing Cys mobile phase concentrations resulted in a proposed mechanism of the toxicological chemistry in blood plasma(Figure 2).

In the SEC experiments, rabbit blood plasma that contained bound CH_3_Hg^+^ was analyzed in the absence and the presence of increasing SMW thiol concentrations in the mobile phase. CH_3_Hg^+^ was fully mobilized from its Cys-34 binding site on RSA using a 50 μM hCys mobile phase, which is the lowest concentration that can be reasonably used with this approach. The results revealed two overlapping Hg peaks which eluted in the SMW elution range (i.e., near the inclusion volume)., One Hg-peak was identified as a CH_3_Hg-hCys complex after repeating the experiment with a 50 mM Tris buffer instead of a PBS buffer and analyzing the corresponding fractions by ESI-MS, while the other Hg peak remained structurally uncharacterized. These findings demonstrate for the first time that CH_3_Hg-hCys can be formed at near-physiological conditions of blood plasma, which is important as this complex and the structurally related CH_3_Hg-hCys have been proposed to interact with L-amino acid transporters 1 (LAT1) at the blood–brain barrier (BBB) [90] for subsequent uptake into the brain [80].

In another SEC experiment, a bis-metalated Hg^2+^ and Cd^2+^-HSA complex (molar ratio 1.0:0.1:0.1) was chromatographically analyzed with a PBS buffer mobile phase, which revealed single co-eluting Cd and Hg peaks [84]. Adding 50 μM Cys to the mobile phase, however, completely mobilized Hg^2+^ from HSA (where it was possibly bound to Cys-34) to form two Hg-bearing metabolites that eluted close to the inclusion volume, while the simultaneously obtained Cd results revealed that only ~50% of this metal eluted as a major protein-bound species followed by three minor Cd-bearing metabolites closer to the inclusion volume. While the employed Cys concentration is higher than that in healthy adults (~10–15 μM), the obtained results nevertheless demonstrate Hg^2+^ to be significantly more ‘mobile’ (i.e., less likely to be bound to HSA) at these near-physiological conditions compared to Cd^2+^. It is highly likely that the high concentration of Cl^−^ in the mobile phase (100 mM) plays an important role in mediating the observed interactions, since several mixed Cl and Cys containing Hg complexes are known to exist [91]. In terms of gaining insight into the structure of the detected Hg and Cd species in the SEC column effluent, it is important to point out that Hg^2+^ and Cd^2+^ are established nephrotoxins (i.e., they damage the kidneys), which suggests that the actual metal species that are uptaken by the kidneys (i.e., the actual nephrotoxic Cd/Hg species) might be among them. Owing to the experimental design, the Hg-bearing metabolite that eluted closest to the inclusion volume (i.e., the one with the larger retention time and thus the smaller species) likely corresponds to (Cys)_2_Hg, which was shown to be organ-available [86]. Interestingly, this tentatively identified (Cys)_2_Hg species co-eluted with the smallest of the SMW Cd species [84]. Evidence in support of the latter Cd species to be a (Cys)_2_Cd complex was obtained by applying AEX to observe its on-column formation in the presence of 100 μM Cys and 100 mM of NaCl [92]. Thus, the co-elution of a Hg and a Cd peak strongly suggests that (Cys)_2_Hg and (Cys)_2_Cd may be the species that are translocated from HSA to the kidneys, possibly involving a shared biomolecular uptake mechanism.

While in vitro results must always be interpreted with caution, they nevertheless suggest that hCys and Cys may play an important role in the RSA-mediated translocation of CH_3_Hg^+^ to the brain and establish an important role for Cys in the translocation of Hg^2+^ [93] and Cd^2+^ from their respective HSA binding sites to the LAT1 uptake mechanism located at the surface of kidney cells (Figure 2). The fact that multiple metal-containing metabolites were observed in these experiments—two for CH_3_Hg^+^ and Hg^2+^ and four for Cd^2+^—suggests that the actual translocation mechanisms, which must involve the uptake of the mobilized metal species into different toxicological target organs may be more elaborate than the mere formation and translocation of just a single metal–SMW thiol complex to an organ. Thus, the known ATP-binding biomolecular ABC-cassette transporters, which are differently expressed on the surface of target organ cells, such as the liver, the kidneys and the lung [94], will determine which of the Cys/hCys containing CH_3_Hg^+^, Hg^2+^ and Cd^2+^ metabolites that are formed in the bloodstream are actually translocated to the toxicological target organs [different transporters will recognize structurally different toxic metal(loid)–Cys/hCys complexes].

Taken together, these results demonstrate that in vitro studies can provide useful insight into dynamic aspects of the toxicological chemistry of individual metal(loid) species in blood plasma and that ternary interactions between a toxic metal(loid) species, a plasma protein and a SMW metabolite (e.g., Cys, hCys and/or others) that unfold in the presence of 100 mM Cl^−^ play an important role in the selective delivery of their ‘cargo’ to toxicological target organs. These findings are reminiscent of previously observed interactions between Cu^2+^, HSA and L-histidine [95], the kinetic aspects of which are still not completely understood [96], and conceptually related studies which have demonstrated that the binding of Zn^2+^ to HSA is modified by the fatty acid myristate, with health-relevant ramifications in the organs downstream [97]. While it cannot be entirely excluded that Hg^2+^-induced conformational changes of HSA [98] play an important role in its translocation to target organs, the obtained results suggest that probing ternary interactions between other neurotoxic metal(loid) species, such as Mn^2+^, As^III^ and Pb^2+^ with their known plasma proteins and distinct SMW metabolites in the presence of 100 mM NaCl may be a fruitful strategy to uncover the biomolecular mechanisms that deliver their cargo to uptake mechanisms at the blood-brain-barrier (BBB), which is a prerequisite for their subsequent uptake into the brain to cause neurotoxicity. Thus, in vitro studies that are conducted at near-physiological conditions appear promising to unravel the biomolecular mechanisms of chronic toxicity that unfold in the plasma–organ system.

## 4. Advances into the Toxicological Chemistry of Metal(loid)s in Red Blood Cells (RBCs)

To gain insight into the binding of Cd^2+^, Hg^2+^ and CH_3_Hg^+^ to cytosolic proteins in rabbit RBCs in vitro, SEC coupled to a multi-element-specific metal detector was successfully applied to probe these interactions for up to 6 h [99]. At the 5 min time point, Cd^2+^ eluted as a major ‘dynamic’ hemoglobin–Cd–glutathione (Hb-Cd-GS) complex (i.e., the observed Cd peak eluted after ‘free’ Hb) and as a minor (GS)_x_Cd complex, while, at the 2 h time point, an additional minor Cd peak eluted implying its binding to an unknown 70 kDa protein. While a minor fraction of both mercurials eluted bound to GSH in the SMW elution range, peaks corresponding to Hg^2+^ and CH_3_Hg^+^ co-eluted with the Fe peak corresponding to Hb [99]. These findings imply a comparatively stronger binding of these mercurials to Hb as compared to Cd at the conditions of RBC cytosol.

The aforementioned Hg results directly relate to recent studies which analyzed blood from fishermen in the Mundaú Lagoon in Maceio, Alagoas (Brazil) who had frequent contact with water and food while working in an area with widespread Hg pollution due to the use of elemental/liquid Hg for the extraction of gold from sediments [100]. The results from these studies revealed elevated concentrations of Hg in the fishermen’s blood (0.73–48.38 μg/L) and urine (0.43–10.2 μg/L) compared with the control group. In addition, the RBCs of these fishermen displayed increased lipid peroxidation (151%), protein oxidation (40%), a reduced activity of antioxidant enzymes [superoxide dismutase (SOD) 26.9% reduction, glutathione peroxidase (GPx) 28.3% reduction and glutathione S-transferase (GST) 19.0% reduction compared with the control group], a 40% reduced O_2_ uptake of Hb and high levels of osmotic fragility, which are indicative of their impaired oxidative status. Thus, the observed in vitro binding of Cd^2+^, Hg^2+^ and CH_3_Hg^+^ to RBC cytosol constituents (i.e., GSH, Hb and other proteins) helps to explain changes in the redox status of the RBCs from fishermen, which, in turn, are implicated in the observed increased prevalence of hypertension and type 2 diabetes (among the 55 fishermen, 34% exhibited hypertension and 11% type 2 diabetes, while in the control group of 25, the corresponding values were 24% and 0%, respectively) [100,101]. Related to these findings, jewelry workers who were occupationally exposed to Cd also exhibited an increased RBC fragility and oxidative stress [102] and red-backed voles living in the vicinity of a metal smelter exhibited similar RBC aberrations [103], clearly demonstrating a need to better understand how bioinorganic chemistry processes of toxic metal(loid) species that unfold in the RBC cytosol cumulatively affect their stability and, thus, the hematological system in more general terms [104]. Closely related to this, it was recently observed that the Zn-metalloprotein carbonic anhydrase I (CA), after it is released from RBCs following their rupture, does not appear to bind to any plasma proteins [105], implicating CA in mediating adverse processes that unfold in the cells that make up the endothelium of blood vessels and are therefore linked to the etiology of atherosclerosis [106]. Interestingly, the exposure of mice to 200 ppb As^III^ in drinking water for 13 weeks resulted in the formation of atherosclerotic plaques, the formation of which was reduced by the co-exposure of mice to a diet containing Se-enriched lentils [107], and there is some evidence that Se-rich lentils, which are classified as a nutraceutical, will also counteract some other adverse effects of As^III^ in humans [108].

## 5. Advances into the Toxicological Chemistry of Cd^2+^, Hg^2+^ and CH_3_Hg^+^ within Target Organs

Since all toxic metal(loid) species/metabolites that are translocated from the blood plasma into target organs will ultimately determine the toxicological effects therein, it is of the utmost importance to identify cytosolic biomolecular targets, which are implicated in unraveling the toxicological mechanisms of action that are involved in mediating organ damage. In addition, it is important to discover biomolecular mechanisms which may be exploited to mobilize toxic metal(loid) species from target organs for their excetion.

To gain insight into the effect of toxic metal(loid) species within toxicological target organs, the analysis of relevant organ cell lysates from environmentally exposed animals with two-dimensional gel electrophoresis (2D-GE) has previously been demonstrated to provide highly useful information about which proteins are up- and downregulated and to identify distinct metalloproteins [109]. The in vivo exposure of numerous mammals to sub-toxic doses of Cd^2+^ has been shown to rapidly induce the gene expression of metallothioneine (MT), which effectively sequesters this toxic metal [110]. Since Cd^2+^ is an established human carcinogen [110], however, other biomolecular mechanisms of action must exist by which this and other toxic metal(loid) species can damage mammalian cells.

In one noteworthy, recent study, rats were injected with Hg^2+^ to deliver a total dose of 8.03 μg/kg over 60 days after which their kidneys were collected. The subsequent homogenization of the kidneys followed by the extraction of the cytosolic proteins allowed to analyze the obtained complex mixture by 2D-GE [111]. While this study revealed 13 upregulated proteins and 47 downregulated proteins compared with the control group, the excision of protein spots followed by their analysis for total Hg by graphite furnace atomic absorption spectrometry revealed 11 Hg-containing spots, some of which contained up to 750 ppm of Hg. Among the identified mercury-protein adducts, there were four unidentified proteins, a major urinary protein, actin isoforms, albumin as well as three Hb proteins, pertaining to Hb subunit beta-1 and Hb subunit beta-2. While the detection of an Hg-albumin complex is not surprising, given that this protein is absorbed from the bloodstream by proximal tubules involving endocytosis [112], it is unclear if the observed Hg– albumin complex was translocated from the bloodstream into the kidneys, possibly involving the already mentioned conformational changes of albumin induced by the binding event [98]. The detection of Hb–Hg complexes in the kidneys, however, is somewhat surprising given that Hb is the most abundant protein in erythrocyte cytosol (~5 mM). It has, however, been reported that Hb is also expressed in mesangial cells in the kidneys [113]. It is still unclear, however, if this Hb–Hg complex was formed within the kidneys (i.e., after Hg^2+^ was uptaken from the bloodstream) or if a Hb–Hg complex that was released after the rupture of RBCs—possibly involving ferroptosis as RBC cytosol contains GPx4 [114]—into plasma and was then translocated to the kidneys, possibly involving endocytosis. Owing to the affinity of Hb to bind to the plasma protein haptoglobin (Hp) it is likely that an Hp–Hb–Hg complex was formed in plasma, which was then endocytosed into the kidneys [115]. The observation of an Hb–Hg complex in the kidneys of rats exposed to Hg^2+^ is of immediate toxicological relevance as free Hb is known to cause kidney damage [116]. Interestingly, chronic kidney disease (CKD), the etiology of which is still unknown, has been demonstrated to induce RBC death, which is referred to as ‘eryptosis’ [117], further underscoring the importance of an integrated approach to establish the biomolecular mechanism of chronic toxicity of potentially toxic metal(loid) species in the bloodstream–organ system.

In terms of identifying the mechanisms by which Cd^2+^, Hg^2+^ and CH_3_Hg^+^ can inflict damage on mammalian cells, two particular mechanisms must be considered. The first one is a distinct form of non-apoptotic programmed cell death called ferroptosis [118], which critically involves iron-mediated Fenton reactions that will eventually cause cell damage/death [119]. Ferroptosis can be triggered by toxic metal(loid) species and has implications for the etiology of neurologic diseases [60]. In brief, ferroptosis involves the inhibition of the cytosolic selenoprotein glutathione peroxidase 4 (GPx4), which contains a selenol (SeH) group and is the only enzyme in mammalian cells which can reduce the cytosolic concentration of lipid peroxides (peroxidized fatty acids), which are produced during cell metabolism. The inhibition of GPx4 will therefore increase the cytosolic concentration of lipid peroxides, which will eventually damage the cell membrane and ultimately kill the cell (Figure 3) [120].

Since the soft acids Hg^2+^ and CH_3_Hg^+^ have a high affinity to bind to SeH groups [121], it was recently reported that the treatment of HEK293T renal cells with these metal species markedly reduced the expression of GPx4 in the nucleus, while the expression of a key regulator protein of ferroptosis PTGS2 was upregulated [122]. These results, combined with the experimentally observed binding of CH_3_Hg^+^ to GPx4, strongly suggest that Hg^2+^ and CH_3_Hg^+^ are able to initiate ferroptosis, thus triggering cell death. Since Cd^2+^ similarly induces ferroptosis in PC12 cells [123], we need to consider that any influx of Hg^2+^, CH_3_Hg^+^ and Cd^2+^ into mammalian cells may induce neurotoxic effects [124], possibly in a cumulative manner. Interestingly, the genetic disruption of GPx4 in the kidneys induced ferroptosis more extensively in males and in females [125]. The second toxicological mechanism of action is the binding of thioredoxin reductase (TrxR) by CH_3_Hg^+^, which was directly observed by X-ray absorption spectroscopy in great detail only recently [126]. This binding event is of toxicological significance as it inhibits TrxR, which plays a vital role in normal cell function [61]. The inhibition of the HSe group of TrxR by Hg^2+^ is a rather complex chain of events which starts with the inhibition of the selenoenzyme, followed by the disruption of the cross-talk between GSH/glutaredoxin 1(Grx1) and thioredoxin (Trx), thus ending in apoptosis mediated via the ASK-1 pathway [127].

The mobilization of toxic metal(loid) species from target organ tissues represents another toxicologically relevant aspect. While orally- or intravenously-administered chelating agents can be an effective strategy to treat As^III^, Cd^2+^, Hg^2+^ and Pb^2+^ exposed individuals [128], adverse effects of chelation therapy frequently occur [129]. Thus, more effective chelating agents that offer fewer side effects continue to be developed [130,131]. Recent results, which involved N-acetyl-L-cysteine (NAC), provided important new insight into the mobilization of CH_3_Hg^+^ from tissues. To this end, it had been phenomenologically observed that the oral administration of mice with NAC dissolved in drinking water effectively mobilized 47–54% of the CH_3_Hg^+^ body burden from the animals within ~2 days [132]. While the relatively low inherent toxicity of NAC, combined with its wide availability in the clinical setting, makes the proposition of orally administering NAC to individuals with a high CH_3_Hg^+^ body burden promising, the bioinorganic mechanism by which NAC actually mobilizes this neurotoxic mercurial from tissues has remained elusive. To unravel this long-standing mystery, a liquid chromatographic approach using a PBS buffer mobile phase which simulated near-physiological conditions (pH 7.4) provided insight into the competitive interaction of Hg^2+^ and CH_3_Hg^+^ with glutathione (GSH) and NAC [133]. Using a C_18_ reversed-phase (RP)-HPLC column, the chromatographic retention behavior of both mercurials was investigated using 5.0 mM GSH in PBS buffer simulating protein-free hepatocyte cytosol. The addition of increasing concentrations of up to 10 mM of NAC to the 5.0 mM GSH in PBS buffer mobile phase resulted in a gradually increasing retention of the Hg peaks corresponding to both mercurials, which implied the on-column formation of more hydrophic complexes. The observed retention behavior at low NAC concentrations was rationalized by the formation of adducts between the hydrophobic NAC with each of the mercurial–GS complexes that were formed with only GSH in the mobile phase (Figure 1, eqn 1–3, left), while, at higher NAC concentrations, the observed increased retention times were rationalize by an eventual displacement of GS moieties that were bound to each mercurial by NAC moieties (Figure 1, eqn 1–3, right).

The observed on-column reaction of Hg^2+^ to form progressively more hydrophobic complexes [GS-Hg-SG → GS-Hg-NAC → Hg(NAC)_2_] and of CH_3_Hg^+^ to similarly form more hydrophobic complexes [CH_3_Hg-SG → CH_3_Hg-NAC] is relevant to understand the bioinorganic process by which NAC mediates the mobilization of CH_3_Hg^+^ from hepatocytes and—owing to the importance of GSH in mammalian cells—possibly in other cell types as well (Figure 4). The results that were obtained for CH_3_Hg^+^ are of particular importance because the observed formation of CH_3_Hg(NAC) at the chosen near-physiological conditions suggest that this complex may be the very Hg species that was excreted from mouse tissues after the oral ingestion of NAC-containing drinking water. In fact, the CH_3_Hg(NAC) complex may represent the species that were excreted from the liver to the bile by a known MRP-2 transporter [134] and from the kidneys by an OAT1 transporter to urine [133].

## 6. A Mechanism-Driven Integrative Framework to Possibly Link Chronic Toxic Metal(loid) Species Exposure of Humans to the Etiology of Environmental Diseases

Every organism needs to regularly exchange nutrients as well as essential and inadvertently toxic ones with its surrounding environment, which will—over its lifetime—determine whether it will remain healthy or eventually develop an environmental disease. Accordingly, the dynamic interactions that unfold after potentially toxic metal(loid) species from the environment enter the GI-tract–bloodstream–organ nexus are of fundamental importance to elucidate which inorganic pollutants are causally linked to the etiology of environmental diseases. This task can be divided into three interrelated problems.

The first and comparatively easy problem is to figure out how an environmentally relevant dose of any given toxic metal(loid) species is being metabolized within humans, what its biological half-life is and by which excretory routes the generated metabolites are excreted (exhaled air, urine, feces; Figure 1). As^III^, for example, is biomethylated to trivalent and pentavalent methylarsenicals, which are rather rapidly excreted in the urine [135], while the biological half-life for CH_3_Hg^+^ in humans is ~50 days [136]. In contrast, there is no physiological mechanism to excrete Cd^2+^ [137]. The second and comparatively more difficult problem is to figure out which toxic metal(loid) species—the parent species and/or its metabolites—perturb which homeostatic regulation mechanisms (HRMs) (Figure 2). To this end, we need to consider that relevant environmental chronic doses of a toxic metal(loid) species may perturb multiple HMRs [138], such as the inhibition of selenoprotein synthesis, the disruption of mitochondrial respiration and/or the induction of ferroptosis (Figure 3), just to name a few. The third and, by far, the biggest problem is to figure out how many HRMs there are (Figure 5) and how we can quantitatively assess the perturbation of each of them to find out to what extent they need to be perturbed to cause the onset of any particular environmental disease [139].

Based on these rather conceptual contemplations, any visual depiction of a mechanism of chronic toxicity for any toxic metal(loid) species must therefore comprehensively describe every biochemical reaction/interaction of a particular metal(loid) species in the bloodstream–organ system with relevant biomolecules (Figure 5) and depict which metabolites are eventually excreted via exhaled air, urine and/or feces. One may therefore classify these interactions—all of which are of toxicological significance—into primary and secondary ones, with the former involving the transport of a toxic metal(loid) species to a biological site (e.g., its binding to a plasma transport protein and/or the formation of a ternary metal-plasma protein–SMW thiol complex for its subsequent uptake into a target organ) and the latter referring to the actual inhibition of a specific target proteins in any given mammalian cell (e.g., ferroptosis).

Although admittedly rudimentary, Figure 5 represents a conceptual depiction of the mechanisms of chronic toxicity for Cd^2+^, Hg^2+^ and CH_3_Hg^+^ in the blood plasma–RBC–organ system. With a focus on addressing the arguably biggest problem in toxicology, namely, multi-pollutant exposure, these bioinorganic processes may be regarded as a framework that may be useful in the design of future experiments toward better understanding the toxicology of environmentally ubiquitous metal(loid) species. Based on this rudimentary framework, four recommendations may be derived to inform the design of future studies: (1) Considering that As^III^, Cd^2+^, Hg^2+^ and CH_3_Hg^+^ are absorbed into RBCs, a focus on detailing their individual interactions with cytosolic constituents is needed, as these can collectively affect their redox status. This may result in RBC fragility and, possibly, ferroptosis-mediated RBC lysis [114], with immediate toxicological ramifications in target organs downstream. (2) There is a need to delineate which plasma proteins and SMW metabolites interact with other toxic metal(loid) species in the presence of 100 mM of Cl^−^ in plasma to direct their toxic cargo to specific biomolecular uptake mechanisms at the surface of toxicological target organs. (3) Another knowledge gap is to better understand the fate of metalloproteins that are released from RBCs (e.g., Hb, carbonic anhydrase 1 and others) in plasma, as these interactions are relevant to adverse effects that unfold within endothelial cells as well as all target organs downstream. (4) Given that many toxic metal(loid) species, owing to their inherent soft acid character, will target similar proteins within mammalian cells (e.g., GPx4, TrxR, the inhibition of DNA-repair processes [140]), the cumulative effect of multiple toxic metal(loid) species, which is the norm rather than the exception, is highly relevant in the context of prenatal exposure which increases the risk of neurodevelopmental impairment among infants [141].

The execution of studies that address these aforementioned problems followed by their integration into a progressively more refined framework will not only advance our understanding of their mechanisms of chronic toxicity, but is also crucial to be able to predict the effect of As^III^, Cd^2+^, Hg^2+^ and CH_3_Hg^+^ on their cumulative perturbation of HRMs. Addressing these knowledge gaps will get us a step closer in our quest to causally link human exposure to multiple toxic metal(loid)s with the etiology of environmental diseases, including neurodevelopment Parkinson’s and Alzheimer’s disease [142,143], autoimmune diseases [144], atherosclerosis [145] and potentially other diseases which do not appear to have a genetic origin [50].

## 7. Conclusions

Due to the enormous economic development and the rapid growth of various anthropogenic activities the insidious environmental exposure of certain human populations, including children and pregnant women, to potentially toxic metal(loid) species in urban centers and near industrial manufacturing sites represents an ongoing public health problem. Understanding whether the increased cumulative influx of these inorganic pollutants into the human bloodstream is tolerable or causally linked to more environmental diseases than we are currently aware of requires a mechanism-driven integrated approach which hinges on the elucidation of bioinorganic chemistry processes involving As^III^, Cd^2+^, Hg^2+^, CH_3_Hg^+^ and Pb^2+^ in the GI-tract–bloodstream–organ system. While the application of synchrotron radiation-based advanced spectroscopic tools have already provided structural information about the chemical nature of Hg and Cd in various human target organs [146,147], our fragmentary knowledge about their individual interactions with biomolecules in blood plasma, RBCs and within organs must be considerably advanced to further detail the mechanisms of the chronic toxicity for individual toxic metal(loids) species as these define—over time—whether the outcome for any given human will be health or disease. To this end, results from studies into the bioinorganic processes that unfold in the bloodstream (e.g., RBC rupture) need to be integrated with those that unfold in target organs (e.g., ferroptosis), as all toxic species that impinge on any given organ and are internalized ultimately determine the resulting organ damage [148]. Therefore, advances in our understanding of these mechanisms will require the structural characterization of novel complexes that are formed between toxic metal(loid) species, plasma proteins and SMW thiols at conditions that resemble blood plasma as this will provide insights into the mechanisms which mediate their uptake into toxicological target organs. Concomitantly, the interaction of toxic metal(loid) species with biomolecules in RBC cytosol needs to be better understood since these are collectively implicated in their lysis, which, in turn, also requires new insights into the biomolecular basis of the toxic effects that the released metalloproteins induce in endothelial cells that line blood vessels, as well as toxicological target organs. The continual refinement of the proposed framework represents an important first step in possibly being able to causally link fundamental bioinorganic chemistry processes that unfold in the bloodstream–organ system with the onset of environmental diseases of unknown etiology. Advances in this research area which will profoundly impact and advance risk assessment, as there is already evidence that the guidelines that are in place to protect human populations from Cd, for example, are inadequate [137]. Analogous to the famous quote by Theodosius G. Dobzhansky that ‘nothing in biology makes sense except in the light of evolution’, one may argue that ‘nothing in the toxicology of metal(loid)s makes sense except in the light of linking exposure to environmental diseases’, as human and planetary health are fundamentally and inextricably interconnected [10,22].

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
