# Peer review of "Toward a Mechanism-Driven Integrated Framework to Link Human Exposure to Multiple Toxic Metal(loid) Species with Environmental Diseases"

_ijms, 2024, doi:10.3390/ijms25063393_

Round 1

Reviewer 1 Report

Comments and Suggestions for Authors

Well structured and nicely written review article. Author covered all relevant sections for the metal(loid) toxicity and their impact in humans as well as their molecular aspects. My specific comments/suggestions are below:

Introduction:

1. 3rd paragraph, 6th line: "in 2015 9 million" should be rephrased.

2. 12th line: "After all, the average life expectancy of Americans began to decline for the first time in 2014 for three consecutive years" Author may rephrased as in previous line author was neither discussing about the life expectancy of any other country nor about any demographic conditions.

Is human exposure.......

3. Toxicity is specifically referred to undesirable concentration of a particular metal(loids). It will be very helpful for readers if author provide a table related to threshold limit of metal(loids) inside human body. Table may also contains natural method to bring down the toxic concentration to below the threshold limit.

 4. After figure 5 2nd paragraph: 11th line, "ramification in tharget organ" must be the target organ.

5. Author may add a section of toxic metal(loid) exposure and government policy, which will address the government regulations which currently under the effect and what are their lacunae and how it will be resolved.

Author Response

I would like to sincerely thank this reviewer for reading the manuscript and the constructive feedback that was provided to improve its clarity.

  1. It was suggested to rephrase ‘in 2015 9 million’ in the introduction, which I have implemented in the revised manuscript.
  2. With regard to changing ‘After all, the average life expectancy of Americans began to decline for the first time in 2014 for three consecutive years’ I thought very hard about rephrasing it, but was unable to come up with a better formulation. I do think, however, that although I just referred to deaths in the sentence preceding this one the reader can understand that the life expectancy can also be affected by the exposure of humans to pollution.
  3. The reviewer requested to provide a Table to highlight the threshold limit of metal(loid)s in the human body. I respond that on page 5 line 1 (revised manuscript) I state that the maximal tolerable daily consumption of As, Cd, Hg and Pb is between 2 and 250 μg per day. I therefore believe that the information this reviewer requested is provided with in the manuscript and I therefore do not see the need to provide an additional Table with this information.
  4. The reviewer requested to correct ‘ramification in tharget organ’, which has been addressed in the revised manuscript.
  5. The reviewer suggested to add a section on the government policies that are in place to address toxic metal(loid) exposures. While I personally think that this does not belong into the manuscript I have provided an additional sentence toward the end of the Introduction (page 10, lines 1-4 in the revised manuscript) in which the interested reader is referred to a recent book chapter that addresses this very issue.

Reviewer 2 Report

Comments and Suggestions for Authors

In this manuscript, the author has attempted to outline an approach for establishing links between transport mechanisms, cellular targets and environmental diseases.

This review is a combination of previously published reviews (for example, https://link.springer.com/article/10.1007/s10534-023-00537-2 and  https://www.sciencedirect.com/science/article/abs/pii/S016201342030307X and so on), but supplemented by several new data obtained in 2023-2024.

In my opinion, the author makes too broad generalizations and makes recommendations for the studies direction, making the manuscript more similar to the type of articles «Viewpoints» (ACS Manuscript Type)

Nevertheless, the manuscript deserves publication in IJMS if several typos and shortcomings are corrected.

1. Figure 1: The excretion of toxic metal(loid) species anD their metabolites in urine, feces and the exhaled air.

Please, correct

2. Please keep the same designation (MeHg+ or CH3Hg+)

3. «While in vitro results must always be interpreted with caution, they nevertheless implicate hCys and Cys in the RSA-mediated translocation of CH3Hg+ to the brain and establish an important role for Cys in the translocation of Hg2+ 91 and Cd2+ from their respective HSA binding sites to the LAT1 uptake mechanism located at the surface of kidney cells (Figure 2).»

This statement is based on the author’s assumptions from other articles and has not been experimentally confirmed. Please rewrite to make it clearer that this is just an assumption.

4. ATB-binding biomolecular ABC-cassette transporter. ATP-binding?

5. «Thus, the known ATP-binding biomolecular ABC-cassette transporters which are differently expressed on the cells surface of target organs will determine which of the Cys/hCys containing CH3Hg+, Hg2+ and Cd2+ metabolites that are formed in the bloodstream are actually translocated to the toxicological target organs.92»

In the cited reference 92 only the transport of cadmium is described and it is unclear why the author believes that this class of transporters will transport other metals.

6. Please, explain in the text what the channel B0+ in Figure 2 is.

7. Explain the abbreviation Hb (on page 9, not on page 10) and GSH (on page 9, not on page 12).

8. «At the 5 min time point Cd2+ eluted as a major ‘dynamic’ Hb-Cd-SG complex (i.e. the observed Cd-peak eluted after ‘free’ Hb) and as a minor (GS)xCd complex, while at the 2 h time point an additional minor Cd peak eluted implying its binding to an unknown 70 kDa proteins»

Author means Hb-Cd-GS complex?

9. Figure 3, please rename LOH as non-peroxidized fatty acids.

10. Please redraw figure 4, since the same figure as TOC is in ref [131]. The same for scheme 1.

11. Figure 4 shows only the mobilization of CH3Hg+ from liver to bile. Please, correct

12. Page 14. This may result in RBC fragility and possibly ferroptosis-mediated RBC lysis, with immediate toxicological ramifications in tHarget organs downstream. Please, correct target

13. Page 15. please replace endothel cells with endothelial cells.

14. «Last but not least, the toxicological chemistry of Cd2+ that unfolds in the GI-tract-bloodstream-organ system will be covered in some detail to critically assess how close we are in terms of establishing the complete sequence of biochemical interactions of this toxic metal species with relevant biomolecules toward establishing a mechanistic framework that may be applied to other metal(loid) species.» (page 5)

I didn't find the toxicological chemistry of Cd2+ that unfolds in the GI-tract-bloodstream-organ system.

Author Response

I greatly appreciate the detailed comments of this reviewer, which I have addressed in the revised manuscript as outlined below.

1.The requested correction was implemented into the revised manuscript.

  1. Thank you very for caching this error. I carefully went through the whole manuscript to make sure the designation for ‘CH3Hg+’ is consistent throughout.
  2. With regard to the passage that this reviewer refers to I have reworded it to denote that the proposed mechanism is an assumption.
  3. I have implemented this error in the revised manuscript.
  4. That you very much for catching this! I have reworded this paragraph to make clear that different ABC-transporters at target organs will mediate the uptake of different metal complexes that are formed in the bloodstream with Cys/hCys.
  5. Thank you very much for pointing this out. I have added a sentence at the end of the figure caption of Fig. 2 and have also revised Figure 2, because it needs to read b0,+ and not B0,+, which is incorrect. I also added a corresponding reference in the caption.
  6. I have introduced the abbreviation of hemoglobin and glutathione sooner in the revised manuscript (i.e. where it is first mentioned).
  7. As requested I have changed ‘Hb-Cd-SG’ to ‘Hb-Cg-GS’.
  8. I have revised Figure 3 to relabel LOH in the figure as suggested.
  9. As requested I redrew Figure 4 in the revised manuscript.
  10. As requested I added the kidney to this figure to denote that the CH3Hg-NAC complex is not only excreted from the liver to bile, but also from the kidneys to the urine.
  11. I corrected this typo (that was mentioned also by reviewer 1) in the revised manuscript.
  12. The requested change was implemented into the revised manuscript.
  13. Thank you for pointing this out. In response I have removed the statement ‘Last but not least…’ (page 5) from the manuscript as I did not address this point in the manuscript.

Round 2

Reviewer 2 Report

Comments and Suggestions for Authors

Well done